

# 1 Climate and soil factors influencing seedling recruitment of

# 2 plant species used for dryland restoration

Miriam Muñoz-Rojas[1,2,3]*; Todd E. Erickson[1,2]; Dylan C. Martini[1,2]; Dixon, Kingsley W.[,1,2,3],
David J. Merritt[1,2]
[1]The University of Western Australia, School of Plant Biology, Crawley, 6009, WA
[2]Kings Park and Botanic Garden, Kings Park, Perth 6005, WA
[3]Curtin University, Department of Environment and Agriculture, 6845 Perth, WA, Australia
(*) Corresponding author at: The University of Western Australia, School of Plant Biology, Crawley, WA
6009, Australia. E-mail addresses: miriammunozrojas@gmail.com, miriam.munoz-rojas@uwa.edu.au.





**Abstract**
Land degradation affects 10-20% of drylands globally. Intensive land use and management, large scale
disturbances such as extractive operations, and global climate change, have contributed to degradation of these
systems worldwide. Restoring these damaged environments is critical to improve ecosystem services and
functions, conserve biodiversity, and contribute to climate resilience, food security, and landscape sustainability
at local, regional and global scales. Here, we present a case study on plant species of the mining intensive semi-
arid Pilbara region in Western Australia that examines the effects of climate and soil factors on the restoration of
drylands. We analysed the effects of a range of climate scenarios (rainfall and temperature) and the use of
alternative soil materials on seedling recruitment of key native plant species from this area. Experimental studies
were conducted in controlled environment facilities where conditions simulated those found in the Pilbara. Air
temperature and soil moisture were modified and monitored routinely. Soil from topsoil (T) stockpiles and
waste materials (W) from an active mine site were mixed at different proportions (100% T, 100% W, and two
mixes of topsoil and waste at 50:50 (TW50:50) and 25:75 (TW25:75) ratios) and used as growth media. Our
results showed that seedling recruitment of the five native plants was highly dependent on soil moisture.
Emergence across the five plant species was higher in the topsoil growth media, which had the highest available
water content compared to the other soil materials. In general, responses to the climate scenarios differed
significantly among the native species which suggest that future climate scenarios of increasing drought might
affect not only seedling recruitment but also diversity and structure of native plant communities. The use of
growth media such as waste materials from mining operations could be an alternative to the limited topsoil.
However, at early plant stages the use of these alternative substrates that are depleted of organic compounds can
be challenging for successful seedling recruitment in the absence of water. These limitations could be overcome
by using soil amendments but the cost associated to these solutions at large landscape scales needs to be
adressed.

**Keywords**
Mine rehabilitation, seedling emergence, native plants, global change, growth media, soil water retention.
**1.  Introduction**
Land degradation affects nearly two billion hectares of land globally, with 25% of the total global land
considered degraded to some extent (Brevik et al., 2015; Stanturf et al., 2015). Restoring these damaged
environments is critical to improve ecosystem services and functions, conserve biodiversity, and contribute to
climate resilience, food security and landscape sustainability at the local, regional and global level (Minnemeyer
et al., 2011; Perring et al., 2015). Drylands, which include semi-arid and arid environments, are particularly
vulnerable to land degradation with estimates suggesting 10-20% of these systems are degraded globally
(Millennium Ecosystem Assessment 2005; Safriel et al., 2005), and continue to be degraded across millions of
hectares every year (Brauch and Spring, 2009). Intensive land use and management, large scale disturbances,
such as extractive operations (e.g. mining), and global climate change have contributed to degradation of these
arid systems worldwide (Kildisheva et al., 2016).
When attempting to restore degraded land, arid ecosystems face the challenges of limited rainfall, high
temperatures, and soils with low nutrient levels and water holding capacity (Anaya-Romero et al., 2015; Muñoz
Rojas et al., 2016a). Thus, despite the efforts and investments to restore these systems worldwide (Keesstra et



al., 2016), restoration of drylands has low rates of success (James et al., 2013; Sheley et al., 2011). To improve
our capacity to reinstate biodiverse, viable plant communities, there is a strong need to advance our
understanding of how these systems function and the effects that environmental and edaphic factors have on
processes such as seedling emergence and plant growth and survival (Perring et al., 2015). For example,
changes in soil water availability as a consequence of reduced rainfall and evaporation, or increases in
temperature due to global warming, may affect restoration outcomes through influencing seedling recruitment
(Cochrane et al., 2015; Lloret et al., 2004) or the composition and distribution of plant species (Lai et al., 2015).
But the impact of environmental factors on restoration can be also compounded by unfavourable edaphic
conditions (Audet et al., 2013; Muñoz-Rojas et al., 2015a). Thus, improving soil physical and chemical
properties can be decisive for successful revegetation (Machado et al., 2013), which is important in extractive
industries operating in dryland environments.
During open-cut and strip mining operations, the top layer of soil is commonly removed and stockpiled before
starting the extraction and then respread before seeding the target sites for restoration (Lamb et al. 2015; Rivera
et al., 2015). This topsoil is an important source of seeds, nutrients and microorganisms (Erickson et al. 2016a;
Golos et al., 2014; Koch, 2007; Muñoz-Rojas et al., 2016b) but its use in restoration is often limited by its
scarcity and the detrimental conditions that topsoil stockpiling can have on soil functionality (Keipert et al.
2002). Waste materials produced in mining operations provide alternative substrates that are currently being
used as growth media in restoration (Machado et al., 2013; Muñoz-Rojas et al., 2016b). These substrates can
integrate coarser materials that help to reduce slope instability and prevent erosion processes, but they are often
highly deficient in organic matter which can reduce soil water retention (Shrestha and Lal, 2006).
Where topsoil is limiting and waste materials form the substrate for plant growth, direct seeding is the most
feasible means of reinstating biodiverse plants communities, particularly at larger scales (Erickson et al. 2016a;
James et al., 2011; Perring et al., 2015; Porensky et al., 2014). However, direct seeding is inefficient in terms of
the proportion of seeds that produce an established seedling; in arid ecosystems it is common for only 2-7% of
seeds to establish (Chambers, 2000; James et al. 2011; Larson et al. 2015). The early developmental life-stages
of plants are usually more sensitive to environmental or edaphic constraints than are the adult stages (Standish et
al., 2014) and the transition from germinated seed to emerged seedling has been identified as the life-stage
transition most limiting the success of direct seeding (James et al. 2011). As these first stages of plant
regeneration fundamentally influence the composition of the future plant community (Jiménez-Alfaro et al.,
2016), characterising abiotic factors of the edaphic environment and their effects on seeds and seedlings is
necessary for developing seeding practices that can achieve the desired restoration outcomes.
With the numerous potential drivers of global change comes a wide range of potential climate change scenarios
(IPCC, 2007). This hinders the incorporation of future climate predictions into restoration programs (Standish et
al., 2014). In this context, more experimental studies are needed to accurately evaluate the effects of altered
climatic conditions on seedling recruitment and subsequent vegetation community structure and function, all of
which, in turn, are strongly linked to soil conditions (Audet et al., 2013). Such experimental approaches can be
effectively addressed by manipulation of combinations of climate and soil factors under controlled conditions
(Lloret et al., 2004; Muñoz-Rojas et al., 2015). Here, we present a case study on plant species of the Pilbara
region in the northwest of Western Australia, where we assess the effects of climate and soil factors on the
restoration of semi-arid ecosystems. The Pilbara (22°03'S, 118°07'E to 23°19'S, 119°43'E) is a vast (179,000
km$^2$) and biodiverse (c. 1800 plant species) semi-arid ecosystem (Erickson et al. 2016a; McKenzie et al., 2009).




The region is subjected to intensive mining, particularly of iron ore, and ecological restoration following mining
commonly requires re-introducing plant propagules to vastly altered growth substrates (Erickson et al. 2016a).
Using five native plant species that form key elements of the vegetation of this ecosystem, the specific
objectives of this study were to: (i) analyse the effects of a range of climate scenarios (rainfall and temperature)
and the use of alternative soil materials on seedling emergence of key native plant species from the Pilbara, (ii)
determine the effects of the climate and soil scenarios on the time to emerge of these plant species, and (iii)
assess the implications of these climate and soil factors on improving the restoration potential in semi-arid
environments.
**2. Methods**
**2.1 Experimental design**
This study was conducted between August and December 2014 in a controlled environment room (CER) at The
University of Western Australia (UWA) and a glasshouse facility at Kings Park and Botanic Garden in Perth,
Western Australia. Five native plant species from five families were selected as representative of a diverse range
of life-forms (e.g. perennial grass, shrub and tree components) that commonly contribute to the mature plant
communities found throughout the mining intensive Pilbara region of Western Australia (Erickson et al. 2016b).
These study species comprised *Acacia hilliana* Maiden (Fabaceae), *Eucalyptus gamophylla* L'Her. (Myrtaceae)*,*
*Gossypium robinsonii* F.Muell. (Malvaceae), *Grevillea pyramidalis* R.Br. (Proteaceae) *and Triodia epactia*
S.W.L.Jacobs (Poaceae). Soil materials commonly used in mine restoration operations in the Pilbara (Bateman
et al., 2016; Muñoz-Rojas et al., 2016b) were collected from an active mining site in the southern part of the
region and used as growth media. These materials consisted of topsoil retrieved from previously stockpiled
material and an overburden waste material commonly used in landform reconstruction due to its erosive stability
and physical competency.
Two experimental studies were carried out to test different climate and soil scenarios. The climate in the Pilbara
region is semi-arid with mean annual rainfall ranging between 250 and 400 mm, mostly concentrated in the
summer months (December to March), accounting for approximately 72% of the total annual rainfall. This
rainfall originates from sporadic summer convection thunderstorms and tropical cyclones. Mean annual
temperatures range between 19.4 and 33.2 °C with average maximums over 40 °C in the summer period
(Bureau of Meteorology, 2015). For the climate scenario experiment we selected a range of precipitation
conditions representative of those of the Pilbara during the summer (growing season) based on the rainfall pulse
duration and the amount of precipitation falling in each event (Bureau of Meteorology, 2015; CSIRO, 2007).
From this selection we developed four simulated rainfall treatments, and a control (e.g. maintained at field
capacity) (Table 1). Three temperature conditions were selected based on daily average temperatures in the
study area (25, 30 and 35°C). These combined rainfall (n=5) and temperature (n=3) treatments resulted in 15
climate scenarios and were evaluated solely in topsoil.
For the soil scenario experiment, a range of growth media blends were evaluated to assess the feasibility of
using growth media mixes in restoration sites. These growth media consisted of four different blends of the soil
materials collected from the mining sites: 100% topsoil (T), 100% waste (W), and two mixes of topsoil and
waste at 50:50 (TW50:50) and 25:75 (TW25:75) ratios. Also, two watering scenarios were set up consisting of a
well watered treatment (WW) and a water deficit treatment (WD). Both treatments were watered 25 ml for 3 d,
then 25 ml every 3[rd] day for WW and every 6[th] day for WD for a total duration of 16 d.



### 2.2. Experimental methods

#### 2.1.1    Soil analyses and measurements

Topsoil and waste material from the mine site were collected and transported to the CER facilities at UWA and Kings Park in 200 l drums. To create each growth media combination, one drum of topsoil (ca. 350 kg) and one drum of waste (ca. 225 kg) were mixed thoroughly into the different blend proportions required (e.g. T, W, TW50:50 and TW25:75, Table 2), ensuring each blend was homogeneous once blended. From each of these growth media blends, three composited soil samples of 500 g were taken, air-dried, and sieved (2 mm mesh) for physical and chemical analysis. Soil pH and electrical conductivity (EC) were calculated in deionised water (1:2.5 and 1:5, w/v, respectively), with a AD8000 microprocessor-based pH. Organic C (OC) was measured by dichromate oxidation (Walkley and Black, 1934) and total N with the Kjeldahl method (Bremner and Mulvaney, 1982). Particle size was analysed by laser diffraction using a Mastersizer 2000 (Malvern Instruments, Malvern, England) after removing the organic matter with $H_2O_2$. Bulk density (BD) was determined according to the method proposed by Rawls (1983).

Soil hydrological parameters (Table 2) were determined according to Conant et al. (2014) using a pressure plate device at four tensions between saturation (-0.001 kPa) and wilting point (-1500 kPa) including field capacity (-10 kPa) (Table 2). Briefly, soil samples were saturated and placed in the pressure plates and then weighed to determine moisture content after hydrostatic equilibrium was reached at each water potential.

#### 2.1.2    Experimental set up

Seeds for each species were obtained from commercially collected seeds supplied to the mining industry for use in Pilbara restoration programs. Upon receipt at Kings Park and Botanic Garden, seeds of *A. hilliana*, *Eucalyptus gamophylla*, *Gossypium robinsonii*, and *Grevillea pyramidalis* were cleaned of any non-seed material (e.g. chaff in *Eucalyptus* collections) and then x-rayed to remove any empty, partially filled, or clearly non-viable seeds (Faxitron MX-20 x-ray cabinet, Tucson, Arizona, USA) following Erickson et al. (2016a). A seed was deemed filled/viable when the x-ray images showed no abnormalities and the image was uniform white/grey in colour. For *Triodia epactia*, a seed is dispersed in an indehiscent floret and requires removal of the floret to maximise the chances of germination (Erickson et al. 2016b). Therefore, seeds were cleaned from the covering florets structures by carefully rubbing florets on a ribbed rubber mat and separating the seed from the floret debris using vacuum separation ('Zig Zag' Selecta, Machinefabriek BV, Enkhuizen, The Netherlands). Seeds were examined under the microscope to ensure no embryo damage occurred. These cleaning processes ensured only > 95% filled/viable material was used in each experiment and removed seed quality as a potential cause of reduced emergence.

To maximise the germination potential of each batch and accommodate seeds with primary dormancy, seed pre-treatments followed pre-treatment recommendations in Erickson et al. (2016a). Seeds of *A. hilliana* and *G. robinsonii* were treated for 1-2 mins at 90°C to break physical dormancy. Seeds of *E. gamophylla* and *G. pyramidalis* were non-dormant and did not require a pre-treatment. Once cleaned from florets, seeds of *T. epactia* were soaked for 24 h in a 1μM concentration of karrikinolide (KAR$_1$; 3-methyl-2H-furo[2,3-c]pyran-2-one, synthesised following Flematti et al. (2005) and re-dried at 15°C / 15% relative humidity for at least 2-3 d prior to sowing.



The climate scenario experiment was conducted in the CER at UWA, where temperature, $CO_2$ and relative
humidity were controlled and monitored routinely. The CER was set to a constant 12 h day and night cycle for
the duration of the experiment, where day-time temperature was the treatment temperature (25, 30 and 35 °C)
and night-time temperature was set at 20° C for all three temperature treatments. Relative humidity was
maintained at 50% and $CO_2$ at 400 ppm. The soil scenario experiment was conducted in the glasshouse facilities
of Kings Park and Botanic Garden where air temperatures where on average 30 °C and relative humidity ca.

178 50%.

For both experiments, pots of 25cm$^2$ surface by 12 cm height were assorted in a randomised block design and
replicated 12 times. Five seeds were sown into each pot and watering regimes were imposed on day 1 of the
experiments and applied manually using a 50ml syringe. Volumetric soil moisture was continuously monitored
across all treatments in three additional 'dummy' pots. An ECHO EC-5 moisture sensor (Decagon Devices,
Inc.) connected to a HOBO micro station data logger (Onset Computer Corporation, Massachusetts, USA) was
inserted completely into the soil surface. Measurements of volumetric soil moisture content were recorded every
5 mins for the duration of the experiment, and were later averaged for daily moisture contents (Fig S1 and Fig
S2). Air temperature was also logged in both experiments.
Seedling emergence was recorded daily in each pot for 16 d. Final emergence (%) was determined as the
average emergence per pot after 16 d divided by five (the number of seeds per pot) and mean emergence time
(MET) was calculated using the following equation adapted from Ellis and Roberts (1980):
$$MET = \frac{\sum Dn}{\sum n} \qquad (1)$$
Where n is the number of seedlings that emerged on day D, and D is the number of days counted from the
beginning of emergence.

*2.1.3    Statistical analyses*
Differences in seedling emergence (final proportion of emerged seedlings among climate and soil scenarios) and
time to emergence among treatments were tested using analysis of variance (ANOVA). Comparisons between
means were performed with the Tukey's HSD (honestly significant difference) test ($P < 0.01$). Before ANOVA
testing, the analysed variables were tested for normality and variance homogeneity using the Shapiro-Wilk and
Levene tests, and data were log transformed as necessary (presented data are non-transformed). All analyses
were performed with R statistical software version 3.1.2 (R Core Team 2014).

**3.    Results and discussion**
**3.1 Climate effects on seedling emergence**
Our results showed that seedling emergence of the Pilbara native plant species was highly dependent on soil
water content in the topsoil growth media (Table 3). Total emergence varied significantly across plant species
and water treatments ($P < 0.001$, Table 3; Fig. 1) and, although we did not find significant differences between
temperature scenarios, interactions of temperature, water and plant species were significantly different ($P <$
0.001, Table 3).



Seedling emergence for *A. hilliana* ranged between 10 and 45% (Fig. 1) and higher values were obtained in the
control and the R1 and R2 treatments (pulse watering treatments of 10 mm and 20 mm daily for 6 d,
respectively). The maximum number of emerged seedlings was recorded at a day temperature of 35°C. Seedling
emergence of *E. gamophylla* followed the same trend with higher emergence in the control, R1 and R2 watering
treatments compared to R3 and R4 watering treatments. For this species, seedling emergence was 20.1±3.8% on
average and up to 40.1 ±6.1% with available water (R1 and R2) and at 35°C. In contrast, emergence of *G.*
*robinsonii* was lower and differences were not significant across water and temperature treatments. Seedlings of
*G. robinsonii* did not emerge at 35°C with short initial pulses of watering (R3 and R4 watering treatments).
However, maximum emergence occurred under this 35°C temperature scenario with the 6-day pulse regime (R1
and R2). Although the maximum seedling emergence recorded for *G. pyramidalis* was higher than the other
species (above 80% in the 30°C scenario), seedlings only emerged with continuous irrigation (control
conditions); suggesting, in terms of seedling emergence, that this species has the lowest tolerance to drought.
Patterns of seedling emergence for *T. epactia* were irregular, but in general, the seeds also proved to be
dependent on higher amounts of water, and emergence generally decreased as temperature increased. Lower
simulated rainfall pulse amounts seemed to be more beneficial for this species (R2 and R4).
Overall, our results showed that seedling emergence of the five native plants studied may decrease in a climate
scenario of increasing drought. However, although rainfall patterns had a large influence on seedling emergence
across all species, changes in temperature did not have such an affect.  These results are broadly consistent with
other similar studies conducted in seasonally dry environments. For example, Lewandrowski (2016) found that
seedling emergence of *Triodia* species decreased as temperature increased. Similarly, in a study of
Mediterranean shrubland of Eastern Spain,  Lloret et al (2004) applied a range of warming treatments with
temperature increments of 0.19-1.12 °C to analyse seedling emergence of native species. They found a moderate
decease in seedling recruitment in the warming treatments compared to the control, but differences were not
statistically significant. Hogenbirk and Wein (1992) obtained larger seedling emergence at higher temperatures,
but only for weedy species, suggesting that climate changes can favour weedy species over native plants. In
general, the climate effect on seedling emergence seems to be more closely connected to water availability than
to warming, and temperature is likely to be less of a limiting factor in the seedling emergence phase for most
species (Lloret et al, 2004; Perring and Hoevenden, 2012; Woods et al., 2010).
In our study, seedling emergence responses to the watering regimes differed significantly among the five
species. We found significantly decreased emergence of seedlings of *G. pyramidalis* and *G. robinsonii* under
water-limited treatments, which suggest that changes in precipitation patterns can have a critical effect on the
recruitment of these species. Plant species producing fewer recruits have been proposed to be more likely to
disappear with drier conditions in future climate scenarios, with a consequent impact on diversity and structure
of native plant communities (Lloret et al. 2004). Thus, the ability of seedlings to make use of the reduced
amount of precipitation for emergence and subsequent survival will be a determinant of their distribution (Lai et
al., 2015).
The mean time for emergence of the five plant species was significantly different across temperature and rainfall
treatments with slightly longer times recorded under higher temperatures; results that are in agreement with
some previous studies (De Frenne et al., 2012; Richter et al., 2012). However, in the southwest of Western
Australia, Cochrane et al. (2015) found that emergence of seedlings was delayed with warmer conditions,



compared to control.  It has been previously suggested that early emergence is a strong determinant of seedling
vigour and can significantly increase plant biomass (Verdú and Traveset 2005).
Regardless of plant species or temperature conditions, our results showed significantly higher rates of emerged
seedlings with longer pulses of simulated rainfall (6 d compared to 2 d) with the same amount of accumulated
water during the treatment (60 ml over the irrigation phase). Semi-arid ecosystems are particularly influenced by
precipitation patterns, and water availability in these environments can be highly pulsed with discrete rainfall
events followed by drought periods (Miranda et al, 2011). Therefore, changes in precipitation frequency, such as
rainfall pulses, can have a stronger effect than rainfall quantity in these environments (Woods et al., 2014).
Another factor that might affect plant production in global climate change scenarios is the elevated
concentration of atmospheric $CO_2$ (IPPC, 2007). However, we have not considered this effect in this study since
it is unlikely that $CO_2$ had any direct impact at the seedling emergence stage (Classes et al., 2010). A number of
studies have previously analysed the possible impacts of $CO_2$ in seedling recruitment but most of them found
that the response of seedling to changes in atmospheric $CO_2$ are constrained by changes in precipitation patterns
(Garten et al., 2008; Kardol et al., 2010).
**3.2 Soil type effects on seedling emergence**
Seedling emergence differed significantly between growth media types, watering treatments and plant species,
but the effect of water inputs seemed to be a larger driver of emergence than growth media type ($P < 0.0001$,
Table 4). With the higher soil moisture treatment (WW treatment), differences between soil materials were not
significant at the $P=0.0001$ level for *E. gamophylla, G. robnsonii* and *G. pyramidalis*, but emergence of *T.
epactia* seedlings was significantly ($P < 0.0001$) higher in the topsoil (56.7± 7.1%) and the 50:50 topsoil:waste
blend (65.1± 7.1%), as compared to the 25:75 topsoil:waste blend (23.3± 6.9%) and the waste (25.1± 5.6%).
Similarly, emergence of *A. hilliana* seedlings showed a progressive decline as the amount of topsoil decreased,
ranging from 58.3 ± 6.3% in the topsoil to 33.3± 7.1% in the waste material. In the WD scenario, seedling
emergence was lower for all species with total emergence varying between 1.7± 1.0 % in *G. pyramidalis* and
40.1±7.1 % in *T. epactia* in the topsoil growth media. In this water limited scenario, seedlings of *G. pyramidalis*
and *G. robinsonii* did not emerge in any growth media apart from the 100% topsoil soil type. Mean time to
emergence did not differ across growth media types (Table 4) or in any of the interactions between growth
media type, water, and plant species.
The analyses of soil physio-chemical properties showed lower contents of sand in the topsoil growth media
(70.5±0.7) consistently increased with increasing fractions of waste in the blend (Table 2). The influence of soil
texture on soil water retention has been largely investigated (Saxton and Rawls, 2006) with different responses
in seedling emergence (Cortina et al., 2011). Soil water holding capacity is generally higher in soils with larger
clay and low sand content (Rawls, 2003). Higher nutrient retention in these soils rich in clay may increase
seedling emergence and seedling root growth, allowing an easier extraction of water from deeper soil profiles
(Woodall, 2010). However, some studies showed that higher infiltration rates in soils with elevated contents of
sand may increase seedling emergence allowing plants to effectively extract water following precipitation
(Cortina et al., 2011).
Our study showed that seedling emergence across the five plant species was higher in the topsoil growth media
which might be explained by the greater water availability as a consequence of larger amounts of organic C
content (Table 2).  Although additional factors, such as adequate nutrient levels in the soil, can be necessary for



plant establishment in degraded soils (Valdecantos et al., 2006; Brevik et al., 2015), water availability seems to
be more critical at early plant life stages, particularly in semi-arid environments (Cortina et al., 2011: Miranda et
al, 2011).

### 3.3. Implications for restoration of degraded lands


The use of suitable growth media such as waste materials has proved to be a competent alternative to the
original soil (i.e. topsoil) in restoration of degraded semi-arid areas (Machado et al., 2013; Muñoz-Rojas et al.,
2015, 2006b; Rivera et al., 2014). Muñoz-Rojas et al. (2016b) showed that soil functions in a rehabilitated area
of northwest Western Australia, with the use of mine waste material, can reach levels of microbial activity and
organic C similar to those of topsoil once vegetation was established. However, here we show that at the early
stages of plant recruitment, the use of alternative substrates depleted of organic materials can be challenging for
successful seedling recruitment in the absence of water. Low contents of soil OC have been commonly
associated to the loss of soil structure, which as a consequence, diminishes water holding capacity, increases
bulk density, and accordingly produces soil compaction (Lal, 2004; Willaarts et al., 2015).
Overall, the results obtained in this study evidence that the availability of water in the soil system is arguably the
most determinant factor for increasing seedling recruitment and, therefore, optimising restoration of semi-arid
lands such as the Pilbara. The application of irrigation has been proposed in restoration of semi-arid systems to
control watering inputs (Bainbridge et al., 2002). There are several types of irrigation systems available that
could effectively increase seedling recruitment, particularly in plant species most sensitive to water limitations
(Padilla et al., 2009). However, there are elevated costs associated to this alternative that makes its use
impractical at the landscape level (Cortina et al., 2011).
Degraded soils – frequently infertile and depleted of organic materials – can respond positively to the addition
of amendments (Cortina et al., 2011; Valdecantos et al., 2006). Soil amendments have been commonly used in
restoration to improve soil structure, restore the hydrological balance and increase the mineral nutritional
capacity (Hueso-González et al., 2014; Jordán et al., 2011). Inorganic amendments (e.g. fertilisers) are usually
applied to overcome plat nutritional deficiencies or physical limitations. However, the use of organic
amendments such as mulch or manure has proved to increase soil water retention in soils with poor structure
with a consequent increase of plant survival in mine restoration (Benigno et al., 2013). Even low doses of
composted organic waste applied in degraded soils have shown to support seedling response for long periods
(Fuentes et al., 2010; Yazdanpanah et al., 2016). Nevertheless, the application of organic amendments can have
several implications such as competition with existing species which is compounded by the high costs of these
practices at large scales in mine restoration (Cortina et al., 2011).
Since seedling establishment from seeds can be challenging in restoration (James et al., 2011), increasing seed
input, or enhancing the availability of suitable micro-sites for seedling emergence through modifying the soil
environment or alternatively improving the regenerative capacity of seeds represent alternative strategies for
those species with limited recruitment (e.g. *G. pyramidalys* or *G.robinsonii*). Such approaches will involve new
technologies for improving seed handling, processing and quality evaluation and the use of seed treatments to
overcome dormancy and improve seedling resilience and vigour germination (Merritt et al. 2007, Turner et al.
2013). For example, though in its infancy, seed coating procedures for native species offer promise of
overcoming recruitment bottlenecks by 'empowering' the seed through coating, pelleting and aggregate
technologies (Madsen et al., 2014; Madsen et al. 2016). Our results highlight the critical impact of soil water



availability for seedling recruitment and the need to address this limitation, but further studies are needed to
develop suitable applications and techniques in drylands restoration at a management scale.  It would be useful
to transfer the experiments reported here to larger-scale field trials to effectively assess applicability of the
findings into restoration programs.
**4. Conclusions**
Our results showed that seedling recruitment of the five native plants was highly dependent on soil moisture and
that temperature did not have a significant effect in the number of emerging seedlings. Emergence across the
five plant species was higher in the topsoil growth media compared to the other soil materials, most likely due to
its larger available water content as a consequence of increased amounts of organic C. Overall, under drought
scenarios total seedling emergence was below 40% for all species and growth media types. In general, responses
to the climate scenarios differed significantly among the five native species suggesting that future climate
scenarios of increasing drought might affect not only seedling recruitment, but also diversity and structure of
native plant communities. In particular, we found significantly decreased emergence rates in seedlings of *G.
pyramidalis* and *G. robinsonii* under water limited treatments meaning that changes in precipitation patterns
may have a critical affect on the recruitment of these species. The use of growth media such as waste materials
from mining operations could be an alternative to the scarce topsoil. However, at early plant stages the use of
these alternative substrates that are depleted of organic materials can be challenging for successful seedling
recruitment in the absence of water. These limitations could be overcome by using soil amendments but the cost
associated to these solutions at large landscape scales needs to be adressed.

**Acknowledgements**
This research was supported by a BHP Billiton Iron Ore Community Development Project (contract no.
8600048550) under the auspices of the Restoration Seedbank Initiative, a partnership between BHP Billiton Iron
Ore, The University of Western Australia, and the Botanic Gardens and Parks Authority.

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



**Tables**

**Table 1**. Simulated rainfall (watering) treatments utilised in this study. Pulse durations and rainfall amounts were selected from interrogating long-term weather data accessed from the Newman Airport weather station (CSIRO 2007; Bureau of Meteorology, 2015). Simulated rainfall treatments (R1 – R4) comprised repeat daily applications of water for either 6 d or 3 d and two different rainfall amounts (20mm or 10mm). The total irrigation amount of 50ml or 25ml matched the pot sizes used in this study and rainfall amount treatments required to simulate the desired simulated rain conditions

| Treatment | Control | R1 | R2 | R3 | R4 |
|---|---|---|---|---|---|
| Pulse duration (days) | - | 6 | 6 | 3 | 3 |
| Rainfall amount (mm) | - | 20 | 10 | 20 | 10 |
| Irrigation (ml) | 50 | 50 | 25 | 50 | 25 |



**Table 2**. Soil physicochemical and hydraulical properties of growth media types (mean ±SE, n=3).EC: electrical conductivity, OC: organic C; N: total N, FC: field capacity, WP: wilting point, AWC: available water content (difference between FC and WP).

| Soil | pH | EC (ms/m) | OC (%) | N (%) | Clay (%) | Silt (%) | Sand (%) | Bulk Density (g/cc) | FC (%) | WP (%) | AWC (%) |
|------|----|-----------|--------|-------|----------|----------|----------|---------------------|--------|--------|---------|
| Topsoil | 7.8±0.1 | 46.7±0.8 | 0.8±0.1 | 0.10±0.01 | 4.6±0.1 | 24.9±0.7 | 70.5±0.7 | 1.55±0.01 | 28.7± 0.2 | 8.9±0.1 | 19.5±0.1 |
| TW (50:50) | 7.6±0.1 | 38.5±2.5 | 0.4±0.1 | 0.03±0.01 | 3.1±0.1 | 21.9±1.7 | 75.0±1.6 | 1.57±0.01 | 19.1±0.4 | 9.1±0.1 | 10.0±0.3 |
| TW (25:75) | 7.8±0.1 | 38.9±2.9 | 0.3±0.1 | 0.02±0.01 | 2.4±0.1 | 12.9±0.7 | 84.7±0.9 | 1.57±0.01 | 17.1±0.3 | 87±0.2 | 8.1±0.2 |
| Waste | 7.3±0.1 | 55.7±10.7 | 0.1±0.1 | 0.01±0.01 | 2.1±0.3 | 11.8±1.0 | 86.0±1.0 | 1.57±0.01 | 12.4±0.4 | 9.1±0.2 | 5.4±0.2 |



1  **Table 3.** Effects of climate factors (temperature and water) and plant species types, and interactive effects of these

2  factors on total emergence and mean time to emerge.  Statistical significance levels: NS: not significant, ***$P <0.001$,

3  **$P <0.01$, * $P <0.05$.

| Factor | Total emergence | | Mean time to emerge | |
|---|---|---|---|---|
| | *F* value | *P* value | *F* value | *P* value |
| Temperature (T) | 2.7802 | NS | 15.5427 | *** |
| Water (W) | 107.5179 | *** | 18.0772 | *** |
| Plant species (P) | 27.9409 | *** | 67.2350 | *** |
| T x P | 3.4951 | ** | 3.2449 | * |
| W x P | 19.6585 | *** | 3.8249 | *** |
| T x W | 2.8951 | * | 0.9380 | NS |
| T x W x P | 3.2669 | *** | 1.3067 | NS |



**Table 4.** Effects of soil or growth media type, water treatments and plant species, and interactive effects of these factors on total emergence and mean time to emerge.  Statistical significance levels: NS: not significant, ***$P < 0.001$, **$P < 0.01$, * $P < 0.05$.

| Factor | Total emergence | | Mean time to emerge | |
|---|---|---|---|---|
| | *F* value | *P value* | *F* value | *P value* |
| Soil (S) | 10.5853 | *** | 0.4043 | NS |
| Water (W) | 301.1846 | *** | 75.6453 | *** |
| Plant species (P) | 19.3987 | *** | 85.6517 | *** |
| S x P | 3.07 | *** | 0.8914 | NS |
| W x P | 12.1949 | *** | 1.3579 | NS |
| S x W | 1.2097 | NS | 0.5689 | NS |
| S x W x P | 3.0291 | *** | 1.9029 | NS |



**Figure captions**
**Figure 1**.Total seedling emergence (%, mean± SE, n=12) of Pilbara native plant species under climate scenarios
(temperature and rainfall). Different letters indicate significant differences over time among watering treatments (C,
R1, R2, R3 and R4) for each temperature scenario (LSD post hoc test, $P < 0.05$). Watering treatments as described in
Table 1.
**Figure 2**. Mean time to emergence (days, mean± SE, n=12) of Pilbara native plant species under climate scenarios
(temperature and rainfall). Watering treatments (C, R1, R2, R3 and R4) as described in Table 1.
**Figure 3.** Total seedling emergence (%, mean± SE, n=12) of Pilbara native plant species for different growth media
types (T: 100% topsoil, TW 50:50: mix of topsoil and waste at 50:50 ratio, TW 25:75: mix of topsoil and waste at
75:25 ratio and W: 100% waste) and watering treatments (WW; well watered and WD: water deficit. Different letters
indicate significant differences over time among watering treatments for each temperature scenario (LSD post hoc
test, $P < 0.01$).



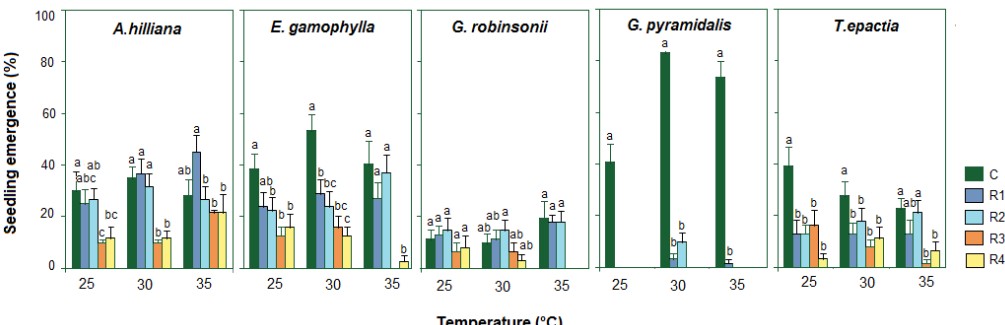


**Figure 1.**Total seedling emergence (%, mean± SE, n=12) of Pilbara native plant species under climate scenarios (temperature and rainfall). Different letters indicate significant differences over time among watering treatments (C, R1, R2, R3 and R4) for each temperature scenario (LSD post hoc test, *P* <0.05). Watering treatments as described in Table 1.






















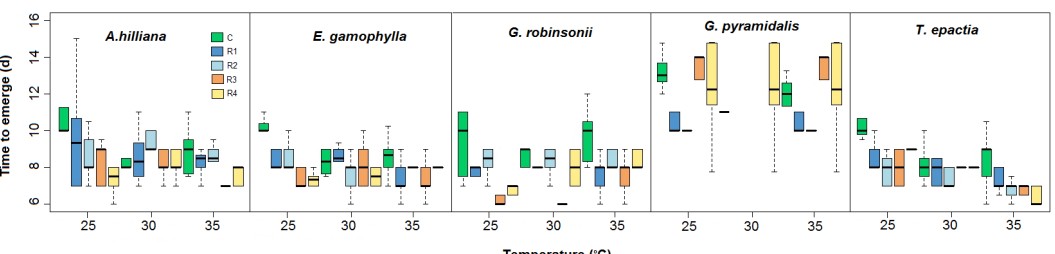


**Figure 2**. Time to emergence (days, mean± SE, n=12) of Pilbara native plant species under climate scenarios (temperature and rainfall). Watering treatments (C, R1, R2, R3 and R4) as described in Table 1.







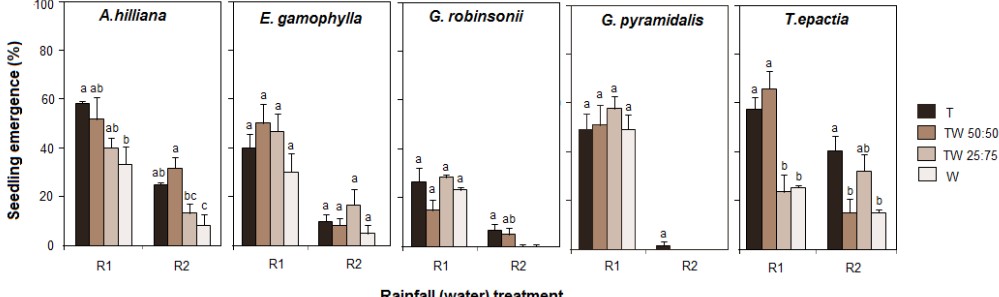


**Figure 3**. Total seedling emergence (%, mean± SE, n=12) of Pilbara native plant species for different growth media
types (T: 100% topsoil, TW 50:50: mix of topsoil and waste at 50:50 ratio, TW 25:75: mix of topsoil and waste at
75:25 ratio and W: 100% waste) and watering treatments (WW; well watered and WD: water deficit. Different letters
indicate significant differences over time among watering treatments for each temperature scenario (LSD post hoc
test, $P < 0.01$).