# Peer review of "Climate and soil factors influencing seedling recruitment of"

_SOIL, 2016_

## Referee Comment (RC1) · Anonymous Referee #1 · 13 Apr 2016

Dear Editor, Dear authors, I found the paper of quality to be published in SOIL I suggest changes in the text and in the figures to make your manuscript consistent, easy to read and attractive to the audience of SOIL See attached my comments Sincerely Artemi Cerdà

**Climate and soil factors influencing seedling recruitment of plant species used for dryland restoration**

Miriam Muñoz-Rojas[1,2,3]*; Todd E. Erickson[1,2]; Dylan C. Martini[1,2]; Dixon, Kingsley W.[1,2,3], David J. Merritt[1,2]

[1] The University of Western Australia, School of Plant Biology, Crawley, 6009, WA

[2] Kings Park and Botanic Garden, Kings Park, Perth 6005, WA

[3] Curtin University, Department of Environment and Agriculture, 6845 Perth, WA, Australia

(*) Corresponding author at: The University of Western Australia, School of Plant Biology, Crawley, WA 6009, Australia. E-mail addresses: miriammunozrojas@gmail.com, miriam.munoz-rojas@uwa.edu.au.

**Fig. 1.**

---

## Referee Comment (RC2) · Anonymous Referee #2 · 9 May 2016

This is an interesting and timely study into factors affecting seedling emergence during restoration of degraded sites in dryland conditions. Overall the experiment and write-up was well-done. I have a few minor comments and suggestions:

Line 61 – I suggest citing Thomas et al., 2015 in addition to the Audet and Muñoz-Rojas citations.

Line 70 – I suggest citing Thomas et al., 2015 in addition to the Machado and Muñoz-Rojas citations.

Line 85 – The 2014 IPCC report is now available, why cite the out of date report?

Lines 185-186 – What are Figs S1 and S2? These were not a part of the file sent to

me. If they are figures meant to be part of a supplement, I suggest incorporating them into the manuscript instead if at all possible/reasonable.

Lines 226-228 – It states here that changes in temperature did not affect seedling emergence. Then it says these results are consistent with similar studies. Then Lewandrowsk (2016) is cited, and it is stated that seedling emergence decreased with increasing temperature in that study. This finding is not consistent with the finding of this study. This section should be reworded/reconsidered as needed.

Lines 246-248 – Here the idea that seedling emergence took slightly longer under higher temperatures is discussed. Then, in the lead-in to the Cochrane et al. (2015) study discussion, the authors begin with "However". Leading with "however" tells the reader you are about to present something that is somehow different than what you have been saying to that point. The Cochrane et al. (2015) study is in agreement with this study. Therefore, the "however" in Line 247 should be removed.

Line 282 – "deeper soil layers" would be more appropriate than "deeper soil profiles". The profile is the soil cross section from top to bottom, and it has layers. We don't have shallow and deep profiles, at any given site we have the soil profile, but we do have shallow and deep layers within a profile.

Line 310 – In addition to the Cortina and Valdecantos references, I suggest citing Lozano-García et al., 2011, and Keesstra et al., 2016 (would become 2016b in this paper).

Line 313 – I think "plat" should be "plant".

Figures 2 and 3 are not called for in the manuscript. A call needs to be added or they need to be deleted.

References Keesstra, S., Pereira, P., Novara, A., Brevik, E.C., Azorin-Molina, C., Parras-Alcántara, L., Jordán, A., Cerdà, A., 2016. Effects of soil management techniques on soil water erosion in apricot orchards. Science of the Total Environment

551–552, 357–366.

Lozano-García, B., Parras-Alcántara, L., Del Toro, M., 2011. The effects of agricultural management with oil mill by-products on surface soil properties, runoff and soil losses in southern Spain. Catena 85, 187-193.

Thomas, C., Sexstone, A., Skousen, J., 2015. Soil biochemical properties in brown and gray mine soils with and without hydroseeding. SOIL 1, 621-629. doi:10.5194/soil-1-621-2015

---

## Author Comment (AC1) · 23 May 2016

We would like to thank Referee #1 for his useful suggestions that we believe have largely improved our manuscript. We also hope that with all these modifications the paper is now suitable for publication in SOIL.

All the suggestions and comments have been addressed, and a new manuscript including updates and modifications is attached. Please, see below some comments in response to the suggestions that Referee #1 made in the pdf document.

- The Abstract has been reduced from 346 to 299 words. - All the citations suggested have been included. These new references have been very useful to support the ideas

and results presented in this paper such as development of new soils in restoration or the challenges of direct seeding. - The figures have been updated (new figures in colour).

Please also note the supplement to this comment:
http://www.soil-discuss.net/soil-2016-25/soil-2016-25-AC1-supplement.pdf

---

## Author Comment (AC2) · 23 May 2016

- We would like to thank Referee #2 for the useful comments that we believe have improved our manuscript. We also hope that with all the modifications the paper is now suitable for publication in SOIL. All the suggestions and comments have been addressed and a new manuscript including updates and modifications is attached. Please, see below a point-to-point response to Referee #2.

This is an interesting and timely study into factors affecting seedling emergence during restoration of degraded sites in dryland conditions. Overall the experiment and write-up was well-done. I have a few minor comments and suggestions:

[Figure]

Line 61 – I suggest citing Thomas et al., 2015 in addition to the Audet and Muñoz-Rojas citations. Line 70 – I suggest citing Thomas et al., 2015 in addition to the Machado and Muñoz-Rojas citations.

- This citation has been added. Please see lines 61 and 70 and the references section.

Line 85 – The 2014 IPCC report is now available, why cite the out of date report?

- This citation has been updated. Please see line 90 and the references section.

Lines 185-186 – What are Figs S1 and S2? These were not a part of the file sent to me. If they are figures meant to be part of a supplement, I suggest incorporating them into the manuscript instead if at all possible/reasonable.

- These figures were uploaded as supplementary material in a separate file. However, they have been incorporated in this new version of the manuscript. Please, see pages 25 and 26.

Lines 226-228 – It states here that changes in temperature did not affect seedling emergence. Then it says these results are consistent with similar studies. Then Lewandrowsk (2016) is cited, and it is stated that seedling emergence decreased with increasing temperature in that study. This finding is not consistent with the finding of this study. This section should be reworded/reconsidered as needed.

- This statement has clarified. Please see lines 229-233: 'Overall, our results showed that rainfall patterns had a large influence on seedling emergence across the five native species, and suggest that seedling recruitment of these native plants may decrease in a climate scenario of increasing drought. These results are broadly consistent with other similar studies conducted in seasonally dry environments. For example, Lewandrowski (2016) found that seedling emergence of Triodia species decreased with water stress and high temperatures (35- 40° C)'.

Lines 246-248 – Here the idea that seedling emergence took slightly longer under higher temperatures is discussed. Then, in the lead-in to the Cochrane et al. (2015)

study discussion, the authors begin with "However". Leading with "however" tells the reader you are about to present something that is somehow different than what you have been saying to that point. The Cochrane et al. (2015) study is in agreement with this study. Therefore, the "however" in Line 247 should be removed.

- Our results actually showed that shorter (and not longer) times recorded under higher temperatures, particularly in A. hilliana and T. epactia. This typographical mistake has been corrected and the paragraph has been modified. Please see lines 250-256: 'The mean time for emergence of the five plant species was significantly different across temperature and rainfall treatments with slightly shorter times recorded under higher temperatures, particularly in A. hilliana and T. epactia (Fig. 2); results that are in agreement with some previous studies (De Frenne et al., 2012; Richter et al., 2012). However, in the southwest of Western Australia, Cochrane et al. (2015) found that emergence of seedlings was delayed with warmer conditions, compared to control. It has been previously suggested that early emergence is a strong determinant of seedling vigour and can significantly increase plant biomass (Verdú and Traveset 2005)'.

Line 282 – "deeper soil layers" would be more appropriate than "deeper soil profiles". The profile is the soil cross section from top to bottom, and it has layers. We don't have shallow and deep profiles, at any given site we have the soil profile, but we do have shallow and deep layers within a profile.

- It has been corrected. Please see line 289.

Line 310 – In addition to the Cortina and Valdecantos references, I suggest citing Lozano-García et al., 2011, and Keesstra et al., 2016 (would become 2016b in this paper).

- These references have been added. Please see line 316 and the references section.

Line 313 – I think "plat" should be "plant".

- It has been corrected. Please see line 319.

Figures 2 and 3 are not called for in the manuscript. A call needs to be added or they need to be deleted.

- It has been corrected. Please see lines 252 and 276.

References Keesstra, S., Pereira, P., Novara, A., Brevik, E.C., Azorin-Molina, C., Parras-Alcántara, L., Jordán, A., Cerdà, A., 2016. Effects of soil management techniques on soil water erosion in apricot orchards. Science of the Total Environment 551–552, 357–366.

Lozano-García, B., Parras-Alcántara, L., Del Toro, M., 2011. The effects of agricultural management with oil mill by-products on surface soil properties, runoff and soil losses in southern Spain. Catena 85, 187-193.

Thomas, C., Sexstone, A., Skousen, J., 2015. Soil biochemical properties in brown and gray mine soils with and without hydroseeding. SOIL 1, 621-629. doi:10.5194/soil-1-621-2015

Please also note the supplement to this comment:
http://www.soil-discuss.net/soil-2016-25/soil-2016-25-AC2-supplement.pdf

**Supplement:**

[revised manuscript text omitted]
versus land-use change by using CORINE Land Cover and MicroLEIS in Southern Spain, International
Agrophysics, 25,395-398, 2011.

Anaya-Romero, M., Abd-Elmabod, S.K., Muñoz-Rojas, M., Castellano, G., Ceacero, C.J., Alvarez, S., Méndez,
M., De la Rosa, D.: Evaluating soil threats under climate change scenarios in the Andalusia region. Southern
Spain, Land Degrad. Develop., 26, 441–449, 2015.

Audet, P., Arnold, S., Lechner, A.M., Baumgartl, T.: Site-specific climate analysis elucidates revegetation
challenges for post-mining landscapes in eastern Australia, Biogeosciences, 10, 6545-6557, 2013.

Bainbridge, D.A.: Alternative irrigation systems for arid land restoration, Ecology Restoration, 20, 23-30, 2002.

Barbero-Sierra, C., Marques, M.J., Ruiz-Pérez, M., Escadafal, R., Exbrayat, W.: How is Desertification
Research Addressed in Spain? Land Versus Soil Approaches. Land Degrad. Develop., 26, 423-432, 2015.

Bateman, A., Lewandrowski, W., Stevens, J., Muñoz-Rojas, M.: The limitations of seedling growth and drought
tolerance to novel soil substrates in arid systems: Implications for restoration success, EGU General Assembly,
Vienna, Austria, 17- 22 April, EGU2016-5557, 2016.

Benigno, S.M., Dixon, K.W. and Stevens, J.C.: Increasing Soil Water Retention with Native-Sourced Mulch
Improves Seedling Establishment in Postmine Mediterranean Sandy Soils, Restoration Ecology, 21, 617–626,
2013.

Bisaro, A., Kirk, M., Zdruli, P., Zimmermann, W.: Global drivers setting desertification research priorities:
Insights from a stakeholder consultation forum. Land Degrad. Develop., 25, 5-16, 2014.

Bochet, E.: The fate of seeds in the soil: a review of the influence of overland flow on seed removal and its
consequences for the vegetation of arid and semiarid patchy ecosystems, SOIL, 1, 131-146, 2015, DOI:
10.5194/soil-1-131-2015.

Brauch, H.G., Spring, U.O.: Securitizing the ground grounding security UNCCD issue paper Nº 2. Secretariat of
the United Nations Convention to Combat Desertification, Bonn, 2009.

Bremner, J.M., Mulvaney, C.S.:. Nitrogen-Total. In: A.L. Page, R.H. Miller (Eds.). Methods of Soil Analysis.
Part 2. 2nd ed. Agron. Monogr. 9. ASA and SSSA, Madison, WI, 595-624, 1982.

Brevik, E.C., Cerdà, A, Mataix-Solera, J., Pereg, L., Quinton, J.N., Six, J., Van Oost, K.: The interdisciplinary
nature of SOIL, SOIL, 1,117–129, 2015.

Bureau of Meteorology, Australian Government. Website accessed 21st of December, 2015, available at
http://www.bom.gov.au/climate/averages/tables/cw_007176.shtml, 2015.

Ceccon, E., González, E.J., Martorell, C.: Is Direct Seeding a Biologically Viable Strategy for Restoring Forest
Ecosystems? Evidences from a Meta-analysis, Land Degrad. Develop., 2015, DOI: 10.1002/ldr.2421

Cerdà, A., García-Fayos, P.: The influence of slope angle on sediment, water and seed losses on badland
landscapes, Geomorphology, 18, 77-90, 1997.

Cerdà, A., García-Fayos, P.: The influence of seed size and shape on their removal by water erosion, Catena, 48,
293-301, 2002, DOI: 10.1016/S0341-8162(02)00027-9

Chambers, J.C.: Seed movements and seedling fates in disturbed sagebrush steppe ecosystems: implications for
restoration, Ecol. Appl., 10, 1400–1413, 2000.

Chaudhuri, S., Mcdonald, L.M., Skousen, J., Pena-Yewtukhiw, E.M.: Soil organic carbon molecular properties:
Effects of time since reclamation in a minesoil chronosequence, Land Degrad. Develop., 26, 237-248, 2015,
DOI: 10. 1002/ldr. 2202

Cochrane, J.A., Hoyle G.L., Yates C.J., Wood J., Nicotra A.B.: Climate warming delays and decreases seedling emergence in a Mediterranean ecosystem, Oikos, 124,150-160, 2015.

Conant, R.T., Dalla-Betta, P., Klopatek, C.C., Klopatek, J.M.: Controls on soil respiration in semiarid soils, Soil Biol. Biochem., 36, 945–951, 2004.

CSIRO: Climate Change in Australia: Tech. Rep. 2007. – CSIRO Publishing, 2007

De Frenne, P. et al.: The response of forest plant regeneration to temperature variation along a latitudinal gradient, Ann. Bot.,109, 1037–1046, 2012.

de Moraes Sá, J.C., Séguy, L., Tivet, F., Lal, R., Bouzinac, S., Borszowskei, P.R., Briedis, C., dos Santos, J.B., da Cruz Hartman, D., Bertoloni, C.G., Rosa, J., Friedrich, T.: Carbon Depletion by Plowing and its Restoration by No-Till Cropping Systems in Oxisols of Subtropical and Tropical Agro-Ecoregions in Brazil, Land Degrad. Develop.,26, 531-543, 2015, DOI: 10.1002/ldr.2218

Drake, J.A., Cavagnaro, T.R., Cunningham, S.C., Jackson, W.R., Patti, A.F.: Does Biochar Improve Establishment of Tree Seedlings in Saline Sodic Soils?, Land Degrad. Develop., 27, 52-59, 2016, DOI: 10.1002/ldr.2374

Ellis, R.A. and Roberts, E.H.: The quantification of ageing and survival in orthodox seeds. Seed Sci. Technol. 9: 373-409, 1981.

Erickson, T.E, Barrett, R.L., Merritt, D.J., Dixon, K.W.: Pilbara seed atlas and field guide: plant restoration in Australia's arid northwest. CSIRO Publishing, Dickson, Australian Capital Territory, 312pp., 2016a.

Erickson, T.E., Shackelford, N., Dixon, K.W., Turner, S.R., Merritt, D.J.: Overcoming physiological dormancy in seeds of *Triodia* (Poaceae) to improve restoration in the arid zone. Restor. Ecol., in press, 2016b.

Flematti, G.R., Ghisalberti, E.L., Dixon, K.W., Trengove, R.D.: Synthesis of the seed germination stimulant 3-methyl-2H-furo[2,3-c]pyran-2-one, Tetrahedron Letters, 46,5719-5721, 2005.

Fuentes, D., Valdecantos, A., Llovet, J., Cortina, J., Vallejo, V.R.: Fine-tuning of sewage sludge application to promote the establishment of Pinus halepensis seedlings, Ecol. Eng., 36, 1213-1221, 2010.

Garten, C.T., Classen, A.T., Norby, R.J., Brice, D.J., Weltzin, J.F., et al.: Role of N2-fixation in constructed old-field communities under different regimes of [CO2], temperature, and water availability, Ecosystems, 11, 125–137, 2008.

Golos, P.J., Dixon, K.W.: Waterproofing topsoil stockpiles minimizes viability eecline in the soil seed bank in an arid environment, Restor. Ecol., 22, 495–501, 2014.

Haigh, M., Reed, H., Flege, A., D'Aucourt, M., Plamping, K., Cullis, M., Woodruffe, P., Sawyer, S., Panhuis, W., Wilding, G., Farrugia, F., Powell, S.: Effect of planting method on the growth of alnus glutinosa and quercus petraea in compacted opencast coal-mine spoils, south wales, Land Degrad. Develop., 26, 227-236, 2015, DOI: 10. 1002/ldr. 2201

Hueso-González, P., Martínez-Murillo, J.F., Ruiz-Sinoga, J.D.: The impact of organic amendments on forest soil properties under Mediterranean climatic conditions, Land Degrad. Dev., 25, 604–612, 2014.

IPCC: Climate Change 2014: Synthesis Report. Contribution of Working Groups I, II and III to the Fifth Assessment Report of the Intergovernmental Panel on Climate Change [Core Writing Team, R.K. Pachauri and L.A. Meyer (eds.)]. IPCC, Geneva, Switzerland, 151 pp., 2014.

[revised manuscript text omitted]

Martín-Moreno, C., Martín Duque, J.F., Nicolau Ibarra, J.M., Hernando Rodríguez, N., Sanz Santos, M.A.,
Sánchez Castillo, L.: Effects of topography and surface soil cover on erosion for mining reclamation: The
experimental spoil heap at el machorro mine (central spain), Land Degrad. Develop., 27, 145-159, 2013, DOI:
10. 1002/ldr. 2232

McKenzie, N., van Leeuwen, S. , Pinder, A.: Introduction to the Pilbara biodiversity survey, 2002-2007, Recs.
Aus. Mus. Supplement, 78, 3-89, 2009.

Miao, L., Moore, J.C., Zeng, F., Lei, J., Ding, J., He, B., Cui, X.: Footprint of Research in Desertification
Management in China, Land Degrad. Develop., 26, 450-457, 2015, DOI: 10.1002/ldr.2399

Millennium Ecosystem Assessment: Ecosystems and Human Well-Being: Desertification Synthesis. World
Resources Institute, Washington, DC, 2005.

Minnemeyer, S., Laestadius, L. and Sizer, N.: A world of opportunity. World Resource Institute, Washington,
DC, 2011.

Merritt, D.J., Turner, S.H.,Clarke, S. and  Dixon, K.W.: Seed dormancy and germination stimulation syndromes
for Australian temperate species, Aust. J. Bot., 55,336–344, 2007.

Muñoz-Rojas, M., Erickson, T., Merritt, D., Dixon, K.: Applying soil science for restoration of post mining
degraded landscapes in semi-arid Australia: challenges and opportunities. EGU General Assembly, 2015,
Vienna, 12-17 April, EGU2015-3967-1, 2015.

Muñoz-Rojas, M., Erickson, T.E., Martini, D., Dixon, K.W., Merritt, D.J.: Soil physicochemical and
microbiological indicators of short, medium and long term post-fire recovery in semi-arid ecosystems, Ecol.
Indic., 63,14-22, 2016a.

Muñoz-Rojas, M., Erickson, T.E., Dixon, K.W., Merritt, D.J.: Soil quality indicators to assess functionality of
restored soils in degraded semi-arid ecosystems, Restor. Ecol., 2016b, DOI: 10.1111/rec.12368.

Padilla, F., Miranda, J., Jorquera, M., Pugnaire, F.: Variability in amount and frequency of water supply affects roots but not growth of arid shrubs, Plant Ecol., 204, 261-270, 2009.

Pallavicini, Y., Alday, J.G., Martínez-Ruiz, C.: Factors affecting herbaceous richness and biomass accumulation patterns of reclaimed coal mines, Land Degrad. Develop., 26, 211-217, 2015, DOI: 10. 1002/ldr. 2198.

Perring, M.P. and Hovenden, M.J.: Seedling survivorship of temperate grassland perennials is remarkably resistant to projected changes in rainfall, Aust. J. Bot., 60, 328–339, 2012.

Perring, M.P., Standish, R.J., Price, J.N., Craig, M.D., Erickson, T.E., Ruthrof, K.X., Whiteley, A.S., Valentine, L.E., Hobbs, R.J.: Advances in restoration ecology: rising to the challenges of the coming decades, Ecosphere, 6, 2015.

Porensky, L.M., Leger, E.A., Davison, J., Miller, W.W., Goergen, E.M., Espeland, E.K., Carroll-Moore, E.M.: Arid old-field restoration: native perennial grasses suppress weeds and erosion, but also suppress native shrubs, Agric., Ecosyst. Environ., 184,135-144, 2014.

Prosdocimi, M., Jordán, A., Tarolli, P., Keesstra, S., Novara, A., Cerdà, A.: The immediate effectiveness of barley straw mulch in reducing soil erodibility and surface runoff generation in Mediterranean vineyards, Sci.Total Environ., 547, 323-330, 2016, DOI: 10.1016/j.scitotenv.2015.12.076

R Core Team: R: A language and environment for statistical computing. R Foundation for Statistical Computing, Vienna, Austria, available at http://www.R-project.org/, 2014.

Rawls, W.J.: Estimating soil bulk density from particle size analyses and organic matter content, Soil Sci., 135,123-125, 1983.

Richter, S. et al.: Phenotypic plasticity facilitates resistance to climate change in a highly variable environment, Oecologia, 169, 269–279, 2012.

Rivera, D., Mejías, V., Jaúregui, B.M., Costa-Tenorio, M., López-Archilla, A.I., Peco, B.: Spreading Topsoil Encourages Ecological Restoration on Embankments: Soil Fertility, Microbial Activity and Vegetation Cover, PLoS ONE , 9(7): e101413, 2014.

Roa-Fuentes L. L., Martínez-Garza C., Etchevers J., Campo J.: Recovery of Soil C and N in a Tropical Pasture: Passive and Active Restoration, Land Degrad. Develop., 26, 201-210, 2015, DOI: 10. 1002/ldr. 2197

Rivas-Pérez, I.M., Fernández-Sanjurjo, M.J., Núñez-Delgado, A., Monterroso, C., Macías, F., Álvarez-Rodríguez, E.: Evolution of Chemical Characteristics of Technosols in an Afforested Coal Mine Dump over a 20-year Period, Land Degrad. Develop, 2016, DOI: 10.1002/ldr.2472

Saxton, K.E., Rawls, W.J.:Soil Water Characteristic Estimates by Texture and Organic Matter for Hydrologic Solutions, Soil Sci. Soc. Am. J., 70,1569-1578, 2006.

Sheley, R.L., James, J.J., Rinella, M.J., Blumenthal, D.M., Ditomasso, J.M.: A Scientific assessment of invasive plant management on anticipated conservation benefits. Pages 291-335 In: Briske DD, (ed) Conservation benefits of rangeland practices: Assessment, recommendations, and knowledge gaps. Allen Press, Lawrence, Kansas, 2011.

Shrestha, R.K., Lal, R.: Ecosystem carbon budgeting and soil carbon sequestration in reclaimed mine soil, Environ. Int., 32,781–796, 2006.

Standish, R.J., Fontaine, J,B., Harris, R.J., Stock, W.D. and Hobbs, R.J.: Interactive effects of altered rainfall
and simulated nitrogen deposition on seedling establishment in a global biodiversity hotspot, Oikos, 121, 2014–
2025, 2012.

Stanturf, J.A., Kant,P., Barnekow Lillesø, J-K., Mansourian, S., Kleine, M., Graudal, L., Madsen, P.: Forest
Landscape Restoration as a Key Component of Climate Change Mitigation and Adaptation, IUFRO World
Series Volume 34. Vienna 72 p., 2015.

Thomas, C., Sexstone, A., Skousen, J.:Soil biochemical properties in brown and gray mine soils with and
without hydroseeding, SOIL, 1, 621-629, 2015.

Torres, L., Abraham, E.M., Rubio, C., Barbero-Sierra, C., Ruiz-Pérez, M.: Desertification Research in
Argentina,  Land Degrad. Develop., 26, 433-440, 2015, DOI: 10.1002/ldr.2392

Turner, S.R., Steadman, K.J., Vlahos, S., Koch, J.M., and Dixon, K.W.: Seed treatment optimizes benefits of
seed bank storage for restoration-ready seeds: the feasibility of prestorage dormancy alleviation for mine-site
revegetation, Rest. Ecol., 21,186–192, 2013.

Valdecantos, A., Cortina, J., Vallejo, V.R.: Nutrient status and field performance of tree seedlings planted in
Mediterranean degraded areas, Ann. For. Sci., 63, 249-256, 2006.

Walkley, A., Black, I.A.: An examination of Degtjareff method for determiningsoil organic matter and a
proposed modification of the chromic acid titrationmethod, Soil Sci., 37, 29–37, 1934

Wang, N., Jiao, J.-Y., Lei, D., Chen, Y., Wang, D.-L.: Effect of rainfall erosion: Seedling damage and
establishment problems, Land Degrad. Develop., 25,565-572, 2014, DOI: 10.1002/ldr.2183

Wang, T., Xue, X., Zhou, L., Guo, J.: Combating Aeolian Desertification in Northern China, Land Degrad.
Develop., 26,118-132, 2015, DOI: 10.1002/ldr.2190

Willaarts, B.A., Oyonarte, C., Muñoz-Rojas, M., Ibáñez, J.J., Aguilera, P.A.:Environmental Factors Controlling
Soil Organic Carbon Stocks in Two Contrasting Mediterranean Climatic Areas of Southern Spain. Land Deg.
Dev., DOI: 10.1002/ldr.2417, 2015.

Yan, X., Cai, Y.L.: Multi-Scale Anthropogenic Driving Forces of Karst Rocky Desertification in Southwest
China, Land Degrad. Develop., 26, 193-200, 2015, DOI: 10.1002/ldr.2209

Yazdanpanah, N.,  Mahmoodabadi, M.,  Cerdà, A.:The impact of organic amendments on soil hydrology,
structure and microbial respiration in semiarid lands, Geoderma, 266, 58-65, 2016,
DOI:10.1016/j.geoderma.2015.11.032

Zucca, C., Wu, W., Dessena, L., Mulas, M.: Assessing the Effectiveness of Land Restoration Interventions in
Dry Lands by Multitemporal Remote Sensing - A Case Study in Ouled DLIM (Marrakech, Morocco), Land
Degrad. Develop., 26, 80-91, 2015, DOI: 10.1002/ldr.2307

**Tables**

[revised manuscript text omitted]